# Perceived discrimination based on the symptoms of covid-19, mental health, and emotional responses–the international online COVISTRESS survey

**Michaël Dambrun**[1]\*, **Eric Bonetto**[2], **Ladislav Motak**[1], **Julien S. Baker**[3], **Reza Bagheri**[4], **Foued Saadaoui**[5], **Hana Rabbouch**[6], **Marek Zak**[7], **Hijrah Nasir**[8], **Martial Mermillod**[9], **Yang Gao**[3], **Samuel Antunes**[10], **Ukadike Chris Ugbolue**[11], **Bruno Pereira**[12], **Jean-Baptiste Bouillon-Minois**[13], **Armelle Nugier**[1], **Maëlys Clinchamps**[13], **Frédéric Dutheil**[13], **The COVISTRESS network**[¶]

**1** Université Clermont Auvergne, CNRS, LaPSCo, F-63000, Clermont-Ferrand, France, **2** Aix-Marseille University, Aix-en-Provence, France, **3** Hong Kong Baptist University, Sport, Physical Education and Health, Kowloon Tong, Hong Kong, **4** Department of Exercise Physiology, University of Isfahan, Isfahan, Iran, **5** Faculty of Sciences, Statistics, King Abdulaziz University, Jeddah, Saudi Arabia, **6** Institut Supérieur de Gestion de Tunis, Université de Tunis, Tunis, Tunisia, **7** Institute of Health Sciences, The Jan Kochanowski University of Kielce, Collegium Medicum, Kielce, Poland, **8** Université Clermont Auvergne, Economic Development, Clermont-Ferrand, France, **9** Univ. Grenoble Alpes, Univ. Savoie Mont Blanc, CNRS, LPNC, Grenoble, France, **10** ISPA—Instituto Universitário, Ordem dos Psicólogos Portugueses, APPsyCI—Applied Psychology Research Center Capabilities & Inclusion, Lisboa, Portugal, **11** Institute for Clinical Exercise & Health Science, University of the West of Scotland, School of Health and Life Sciences, South Lanarkshire, Scotland, United Kingdom, **12** CHU Clermont-Ferrand, Biostatistics, Clermont-Ferrand, France, **13** Université Clermont Auvergne, CNRS, LaPSCo, Physiological and Psychosocial Stress, CHU Clermont-Ferrand, WittyFit, Clermont-Ferrand, France

¶ Members of the COVISTRESS network are provided in the Acknowledgments
\* michael.dambrun@uca.fr

## Abstract

### Background

Despite the potential detrimental consequences for individuals' health and discrimination from covid-19 symptoms, the outcomes have received little attention. This study examines the relationships between having personally experienced discrimination based on the symptoms of covid-19 (during the first wave of the pandemic), mental health, and emotional responses (anger and sadness). It was predicted that covid-19 discrimination would be positively related to poor mental health and that this relationship would be mediated by the emotions of anger and sadness.

### Methods

The study was conducted using an online questionnaire from January to June 2020 (the Covistress network; including 44 countries). Participants were extracted from the COVISTRESS database ($N_{total}$ = 280) with about a half declaring having been discriminated due to covid-19 symptoms ($N$ = 135). Discriminated participants were compared to non-discriminated participants using ANOVA. A mediation analysis was conducted to examine the

**Data Availability Statement:** Data Availability Statement Dambrun, Michael (2022): Dataset: Perceived discrimination based on the symptoms

of covid-19, mental health and emotional responses - the international online COVISTRESS survey. Figshare. Dataset. 10.6084/m9.figshare.21103405.

**Funding:** The authors received no specific funding for this work.

**Competing interests:** The authors have declared that no competing interests exist.

indirect effect of emotional responses and the relationships between perceived discrimination and self-reported mental health.

## Results

The results indicated that individuals who experienced discrimination based on the symptoms of covid-19 had poorer mental health and experienced more anger and sadness. The relationship between covid-19 personal discrimination and mental health disappeared when the emotions of anger and sadness were statistically controlled for. The indirect effects for both anger and sadness were statistically significant.

## Discussion

This study suggests that the covid-19 pandemic may have generated discriminatory behaviors toward those suspected of having symptoms and that this is related to poorer mental health via anger and sadness.

## Introduction

A significant body of work reveals that the covid-19 pandemic has negatively impacted the mental health of populations [1], although there is now some evidence of resilience over time with a return to normality [2]. While some factors have been identified as having a massive role on distress during the pandemic (e.g., social isolation, perception of health risks associated with the virus; [3]), other variables have received little attention as they are likely limited to a smaller number of individuals. This is the case, for example, for social rejection and discrimination due to covid-19 symptoms [4,5]. While direct empirical evidence remains scarce, covid 19-related discrimination may occur through relatively well-known mechanisms.

Social rejection and discrimination (i.e., the subjective experience of being rejected and treated unfairly relative to others, due to membership of a category) has been established as an important psychological stressor [6]. Perceived discrimination has been found to be associated with a variety of health issues, such as high blood pressure [7,8], poorer psychological and physical health [9], low levels of life satisfaction and well-being [10,11], high levels of psychological distress and depressive symptoms [6,12–14], high levels of obsessive-compulsive symptoms [15], poorer cognitive test performance [16], and poor sleep quality [17], to name a few. Moreover, when discrimination is perceived to be related to individual-specific, uncontrollable, and persistent characteristics, it can promote feelings of helplessness, hopelessness, and depressed mood status [18,19].

Covid-19 related discrimination is no exception. When confronted with an individual showing symptoms consistent with covid-19 (e.g., coughing, blowing nose), the behavioral immune system promotes behaviors that prevent contact with objects and individuals potentially carrying the threat [20]. This can lead to simply avoiding contact with the individual in question (e.g., social distancing; [21]), but it can also lead to behaviors of implicit disapproval (e.g., disparaging looks from others, suspicion) or more explicit disapproval (e.g., verbal aggression), social rejection, or even physical violence [22]. Such discriminations, going beyond the simple avoidance of the pathogen threat, have been observed in previous viral outbreaks, such as severe acute respiratory syndrome (SARS) and Ebola [23–25].

Previous studies have indicated that covid-19 sufferers have experienced similar responses, with about one-fifth of the surveyed samples reported having experienced discrimination (e.g., 22.10% in a US sample, [26]; 18% in an Indian sample, [22]), without including the potential of under-reporting of such phenomena (respondents preferring not to answer questions related to such discriminations). Moreover, studies have shown positive relationships between perceived covid-19 discrimination and various mental health problems. Perceived covid-19-related discrimination has been found to be positively associated with psychological distress [27], anxiety and depression [28]. Consequently, understanding and fighting discriminations based on the symptoms of covid-19 constitutes a health priority [29,30].

In this study, we observed discrimination towards individuals associated with covid-19, especially those showing symptoms consistent with the disease [29,30]. We examined the relationship between perceived discrimination based on covid-19 and mental health. In addition, we explored emotional responses to perceived discrimination as an explanation for this relationship. Specifically, we examined the extent of how the emotions of anger and sadness were related to the experience of discrimination and how they could explain the relationship between perceived discrimination based on covid-19 and mental health.

Experiencing discrimination may be associated with various emotions that impair both mental health and well-being. Anger is particularly likely when discrimination is perceived as unfair [31,32]. Using anger to cope with discrimination has been found to negatively impact general well-being and psychological distress [33]. In addition, anger is positively associated with a variety of health issues such as high blood pressure [34] and coronary heart disease outcomes [35]. For its part, sadness constitutes a common emotional response to perceived discrimination [36]. Sadness has also been found to be associated with various mental health issues (e.g., bipolar disorders, anxiety, psychotic disorders, suicide attempts; [37]).

Therefore, we predict that perceived discrimination based on covid-19 symptoms will be negatively associated with mental health (i.e., anxiety, stress, burnout, low self-esteem). In addition, perceived discrimination will be positively associated with anger and sadness. Finally, we predict that these emotional responses will mediate the relationship between perceived discrimination and low mental health.

## Method

### Study design

In this study we used an anonymous online computerized questionnaire, accessible via the COVISTRESS.ORG website. The questionnaire was translated into ten languages. The questionnaire was hosted by Clermont-Ferrand University Hospital, using REDCap® software. To facilitate dissemination, the questionnaire was distributed using different methods (social networks, radio, television, internet, mailing lists, etc.). Respondents were informed of the purpose of the survey before completing the questionnaire. They were also informed that their data would be used anonymously for research purposes. The study was approved by the South East VI Ethics Committee of France (Clinicaltrials.gov NCT04538586). The ethics committee waived the need for written consent considering that the respondents gave their consent by answering the questionnaire. Respondents could also withdraw it at any time.

### Participants

Participants were extracted from the COVISTRESS database in January 2021 (the questionnaire was disseminated from January to June 2020; [38]). At the time of the extraction, there were 13429 participants in this database ($M_{age}$ = 41.03, $SD$ = 13.97; 60.90% of female). The COVISTRESS questionnaire was disseminated to the general population in 44 countries. The distribution of responses

by continent was as follows: Europe 82.8%, America 8.5%, Africa 5,2%, Asia 3.4% (for more details, [38]) without gender distinction, occupation or disease. As shown in Fig 1, we then extracted the participants who indicated that they were discriminated based on symptoms consistent with covid-19 ($n$ = 135; 1%). Finally, we randomly selected 145 participants who did not perceive discrimination to establish a comparison group. Equivalence analyses of these two groups on several relevant variables were performed to ensure comparability between the two groups (Table 1). The final sample size for this study was 280 participants ($M_{age}$ = 41.38, $SD$ = 12.59; 73.20% of female).

## Material and procedures

The administered survey was a computerized questionnaire hosted on the secure REDCAP® platform. It was composed of about 100 questions. The study relied on the answers relating only to the variables presented.

The main outcome measure was the perceived discrimination, based on symptoms consistent with covid-19. The participants were asked if they had experienced discriminatory reactions from others. If so, they were asked to specify what their perception of discrimination was based on. Among the possible choices in relation to the symptoms consistent with covid-19 (at the time of the first covid-19 wave), there were "*I coughed*", "*I blew my nose*", "*I wore a mask*". If the participants checked any of these boxes, they were considered to have experienced covid-19-based discrimination. Supplementary questions regarding the type, the perceived impact and the context of discrimination were available, and participants could select the answers of their choice (Table 2).

Furthermore, emotional responses were assessed using two visual analog scales (VAS), one for anger and one for sadness. Ranging from 0 (minimum) to 100 (maximum), these scales allowed participants to express to what extent they felt angry (100, vs. peaceful, 0) or sad (100,

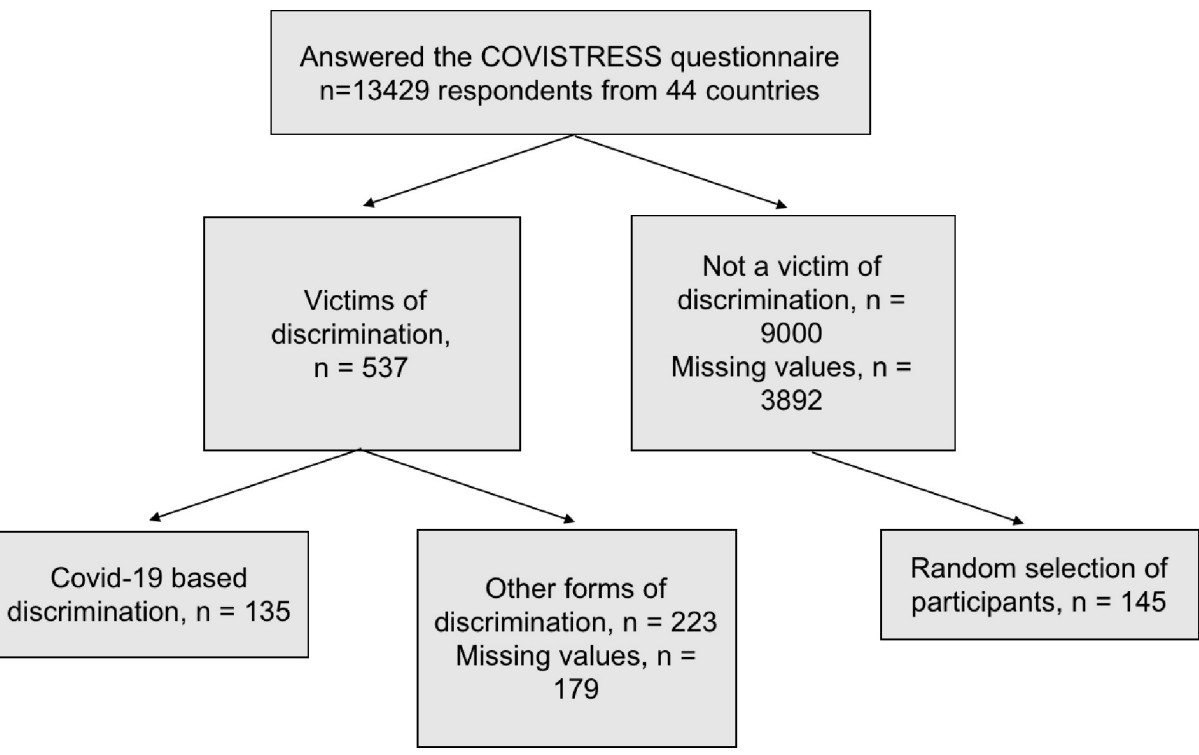

**Fig 1. Flow chart.** Recruitment characteristics of the study cohort.

**Table 1. Mean values (and standard deviations) for the main participants' characteristics between the two groups (no- vs. discrimination).**

| Variable | Discrimination = yes (*n* = 135) | Discrimination = no (*n* = 145) | *p*-value |
|---|---|---|---|
| Sex (M/F Ratio) | 29/77 | 34/107 | 0,75 |
| Age | 39.53 (12.2) | 41.22 (12.9) | 0,27 |
| Occupation | | | 0,76 |
| . Student | 7% | 8% | |
| . Looking for a job | 8% | 11% | |
| . Executive and superior | 31% | 33% | |
| . Intermediary | 35% | 29% | |
| . Merchant, business | 9% | 6% | |
| . Retiree | 4% | 5% | |
| . Worker | 6% | 8% | |
| . Farmer | 1% | -- | |
| Education | | | 0,84 |
| . PhD or other Master+ | 9% | 12% | |
| . High school graduates | 14% | 12% | |
| . 1st to 2nd university yr | 39% | 40% | |
| . 5th yr university level (M2, MDE, DHS or DHAS) | 30% | 25% | |
| . Youth training / BTEC | 8% | 10% | |
| .GCSE or under | 1% | 1% | |
| Marital status | | | 0,69 |
| . Single | 29% | 29% | |
| . Widowed | 2% | 1% | |
| . Cohabitation | 15% | 9% | |
| . Married | 33% | 40% | |
| . Other | 22% | 20% | |
| Number of children | 1.11 (1.2) | 1.23 (1.3) | 0,43 |
| Smoking (cigarettes per day) | 2.82 (6.0) | 2.89 (5.7) | 0,93 |
| Alcohol consumption (drinks a week) | 5.93 (6.3) | 6.82 (6.1) | 0,27 |
| Cannabis consumption (joints a week) | 0.41 (2.4) | 0.49 (2.2) | 0,80 |
| Contact with covid-19 (0 = no; 1 = yes) | | | 0.12 |
| . No | 15,60% | 21.9% | |
| . Yes | 25.9% | 16.8% | |
| . I don't know | 58.5% | 61.3% | |
| Antidepressant use (0 = no; 1 = yes) | | | < 0.05 |
| . No | 85.9% | 93.1% | |
| . Yes | 14.1% | 6.9% | |

vs. joyful, 0). Such scales are validated tools commonly used in daily practice to assess emotions and mental health [39,40]. Mental health was the third outcome measure using VAS relating to stress at home; stress at work; fatigue; burnout; and self-esteem (six items, Cronbach's $\alpha$ = .77). Finally, secondary outcomes were socio-demographic (age, gender, occupation, education, marital status, number of children) and various consumption variables (smoking, alcohol and cannabis use and antidepressant treatment).

## Statistical analyses

Data were expressed in number and percentage for categorical variables and as mean ± standard deviation (SD) for quantitative variables. Statistics were computed using

**Table 2. Frequencies concerning type of discrimination, perceived impact and the context of discrimination (discrimination group only).**

| Type of discrimination | % |
|---|---|
| Disparaging looks from others | 82.2 |
| Imposed isolation | 11.9 |
| Verbal aggression (insults) | 23.0 |
| Physical assault (hitting) | 0.0 |
| Perceived impact | % |
| I ignored it and it didn't affect me | 45.9 |
| I pretended to ignore it but it affected me | 32.6 |
| It affected me moderately | 25.2 |
| It affected me strongly | 8.9 |
| Context of discrimination | % |
| At work | 25.2 |
| At the hospital or place of care | 8.9 |
| My home | 5.2 |
| In the street | 45.2 |
| Public transportation | 8.9 |
| Supermarket/shopping | 48.1 |
| Other public place | 11.9 |

Jamovi software (Version 2.3.3) and SPSS 25 (Version 25). Comparisons between categorical variables were accomplished using Chi2 ($\chi^2$) and contingency tables. Comparisons between quantitative variables, such as age, number of children, smoking, alcohol, and cannabis consumption, were executed using ANOVA (Table 1). Comparisons between mental health and emotional responses were executed using ANCOVA with antidepressant use as a covariate (Table 3 and Fig 2). A Pearson's r test was carried out to investigate the correlation between emotional responses and mental health. Finally, a mediation analysis was conducted using Process 3.5 for SPSS (95% *CI*; Number of bootstrap samples = 5000) to examine the indirect effect of emotional responses in the relationship between perceived discrimination and self-reported mental health (Table 4 and Fig 3). As before, antidepressant use was added as a covariate.

## Results

### Matching people that perceived discrimination with those that did not have perceived discrimination

We checked that there were no differences between our two groups using several controlled variables (Table 1). The only difference between the two groups was the use of antidepressants, which was significantly higher in the group that perceived discrimination. Therefore, this variable was systematically added as a covariate in the statistical analyses comparing the two groups.

### Type of discrimination, perceived impact, and the context of discrimination

Most of the discriminated individuals reported having been subjected to derogatory looks from others. One in four had been verbally insulted and one in eight had been placed in isolation solely because of a symptom (Table 2). Regarding the perceived impact, 45.9% said they

**Table 3.  Marginal mean estimates (and standard deviations) for both mental health and emotional responses, within both no- and discrimination groups, when depression medication is controlled for (analysis of covariance).**

|  | Discrimination = yes (n = 135) | Discrimination = no (n = 145) | $F$ | Eta$^2_p$ |
|---|---|---|---|---|
| Poor mental health | 59.3 (20.9) | 49.8 (21.1) | 9.02** | 0.05 |
| • Stress at home | 55.8 (32.3) | 45.6 (34.1) | 5.08* | 0.02 |
| • Stress at work | 63.1 (35.4) | 54.9 (37.2) | 3.40 | 0.01 |
| • Fatigue | 65.8 (29.5) | 56.1 (31.0) | 5.49* | 0.02 |
| • Anxiety | 60.7 (28.1) | 53.7 (32.5) | 2.80 | 0.01 |
| • Burnout | 51.9 (33.7) | 38.4 (30.7) | 9.76** | 0.04 |
| • Self-esteem | 45.1 (27.4) | 55.4 (27.8) | 6.12* | 0.03 |
| *Emotional responses* |  |  |  |  |
| Anger | 67.5 (24.3) | 56.2 (28.8) | 9.05** | 0.04 |
| Sadness | 61.7 (27.0) | 52.8 (26.1) | 5.54* | 0.03 |

*Note:* ** $p < .01$,

* $p < .05$.

ignored discrimination and were not affected. One in four said they were affected moderately, 8.9% strongly, and 32.6% pretended to ignore the discrimination but said they were affected. The contexts are varied: 48.1% say they were discriminated against in the supermarket while shopping, 45.2% in the street, and 25.2% at work. No such patterns were reported by members of the non-discrimination group.

## Covid 19-based discrimination impacts on self-reported mental health and on emotional responses

An ANCOVA (controlling for antidepressant use) on the composite scores of mental health revealed a significant main effect of perceived discrimination (Table 3). Participants who experienced discrimination reported poorer mental health ($M = 60.8$) than those who did not ($M = 51.4$). The analysis for each symptom taken separately revealed significant effects for stress at home, fatigue, burnout, and self-esteem.

Similar analyses on the scores of anger and sadness also revealed significant main effects of perceived discrimination. Those who experienced discrimination because of covid-19 symptoms felt significantly angrier and sadder (respectively, $M = 67.5$, SD = 24.3 and $M = 61.7$, SD = 27.0) than those who did not experience discrimination (respectively, $M = 56.2$, SD = 28.8 and $M = 52.8$, SD = 26.1; Table 3 and Fig 2).

## Do anger and sadness explain the relationship between perceived discrimination and mental health?

Firstly, we examined the Pearson correlations between emotional responses and mental health. While anger and sadness were correlated positively and significantly to each other ($r = .51$, $p < 0.001$), both were significantly related to poor mental health ($r = .37$, $p < 0.001$ in the case of anger, and $r = .38$, $p < 0.001$ in the case of sadness). We examined the indirect effect of emotional responses in the relationship between perceived discrimination and self-reported mental health. As before, antidepressant use was added as a covariate.

Consistent with previous results, the total effect was significant (Table 4). The direct effect became non-significant when both anger and sadness were statistically controlled for. Both the indirect effect of anger and sadness were statistically significant, indicating that both mediated independently the relationship between perceived discrimination and mental health (Fig 3).

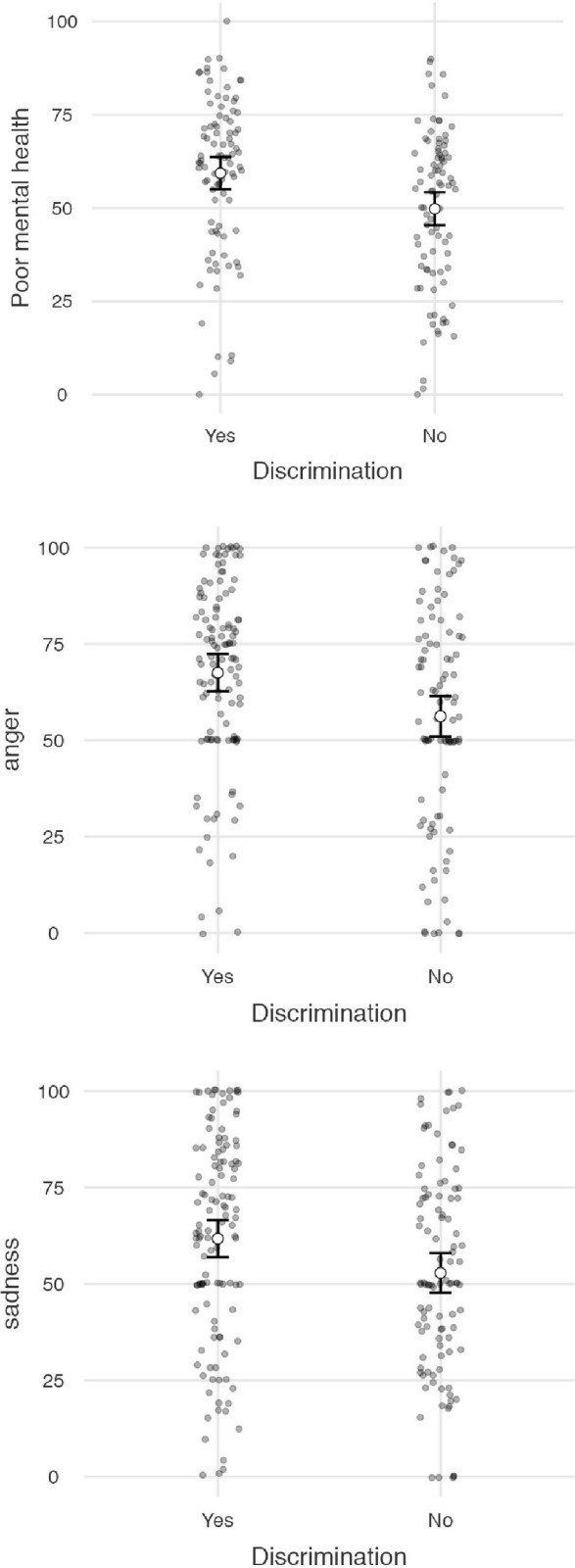

**Fig 2. Marginal mean estimates (and confident intervals) for both mental health and emotional responses, within both no- and discrimination groups.**

**Table 4. Total, direct and indirect effects on the relation between perceived discrimination (X) and mental health (Y) via emotional responses (i.e. anger and sadness).**

|  | *b* | *se* | *p*-value | 95% *CI* |
|---|---|---|---|---|
| Total effect of X on Y | -9.78 | 3.34 | 0.004 | -16.39, -3.17 |
| Direct effect of X on Y | -5.98 | 3.17 | 0.061 | -12.25, 0.29 |
| Indirect effects: | | | | |
| • Anger | -1.80 | 1.23 | | -4.72, -0.07 |
| • - Sadness | -2.00 | 1.12 | | -4.57, -0.23 |

## Discussion

When confronted with an individual showing symptoms consistent with covid-19, the behavioral immune system promotes behaviors that prevent contact with objects and individuals potentially carrying the threat [20]. This can lead to discriminatory attitudes or behaviors [29,30]. However, such discriminative behaviors tend to be detrimental to the mental health of the targeted individuals [27,28]. The present research replicates this detrimental effect of perceived discrimination on individuals' mental health. More precisely, our results show that covid-19 discrimination seems to be positively related to poor mental health. Although subtle in their effect sizes [2], stronger levels of stress and fatigue, blatantly higher levels of burnout, and lower self-esteem are more salient within individuals who did report covid-19 discrimination than within their non discriminated counterparts.

The present research further explored the detrimental effect of discrimination on mental health through the investigation of the role of negative emotions in this relationship. As expected, this relationship appeared to be mediated by the emotions of anger and sadness. This last result is in line with the literature showing that using anger to cope with discrimination tends to favor psychological distress [33], and that individuals may commonly respond to discrimination with both sadness [36] and anger [41].

The present research may have implications for public health strategies. Our results parallel previous work showing that emotion-focused coping is related to perceived discrimination and mental health in other social contexts [42,43]. Without taking the place of problem-focused coping strategies in dealing with psychological consequences of perceived discrimination (a particularly efficient strategy; [44]), interventions should also aim to improve more positive emotional coping strategies among discriminated individuals to reduce negative emotions, and consequently promote mental health. Moreover, previous studies showed the positive effects of emotional approach coping in the context of various stressors (e.g., breast cancer, chronic pain; [45,46].

As stated above, emotions are supposed to mediate the link between discrimination on one hand and mental health issues on the other [33,36,47,48]. One promising line of public health strategy may emphasize positive emotions which–for example, when "savored",–try to enhance positive emotion regulation [49]. Such an enhanced regulation leads to a more functional reappraisal [50] and may, in turn, help in resisting discrimination [47,51,52].

The results of the present study remain constrained by several limitations. This study was conducted in 44 countries using different languages. It is possible that there were cultural variations that have not been identified here and may be included when the covistress cohort is of a sufficient size in each country to enable inclusion. Second, the present research replicated the effects of covid-19 related discrimination on self-reported mental health. To this end, we used a composite measure of mental health (i.e., stress at home, stress at work, anxiety, fatigue, burnout, self-esteem) that displayed good internal consistency. However, it is interesting to note that no difference was observed between participants who perceived discrimination and

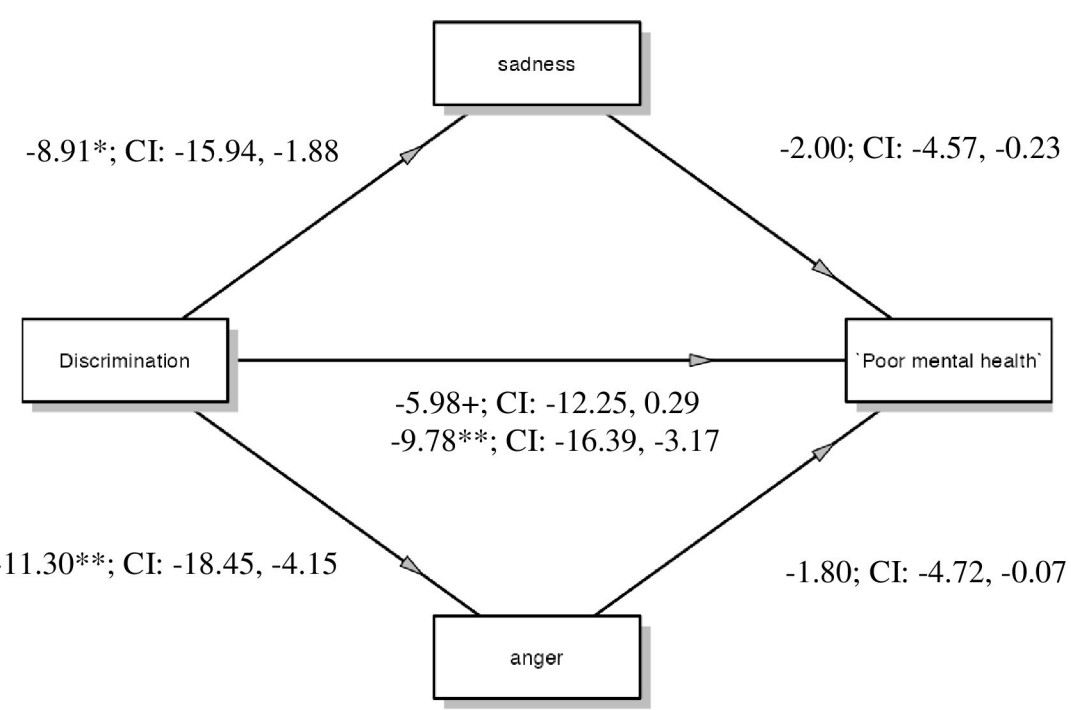

**Fig 3. Multiple mediation model: Anger and sadness mediate the relationship between covid-19 based discrimination and poor mental health.** *Note*: ** $p < 0.01$, * $p < 0.05$, + $p < 0.10$.

those who did not in terms of stress at work and anxiety. Yet, perceived discrimination related to covid-19 symptoms is often experienced at work (Table 2), unemployed individuals experiencing less covid-19 discrimination (as reflected in the greater tendency to burnout among those who experienced such discrimination; Table 3) and tends to cause anxiety in individuals experiencing it [28]. Future research will need to investigate the label of "mental health" to study the differential effects of perceived discrimination on the different facets of this construct. Future research should also use more relevant physiological responses in the context of discrimination [7,53]. Third, the discrimination measure used was of the binary type, therefore not very sensitive, but sufficiently so to detect an effect on mental health and emotions. It would be relevant to replicate this study with a likert-type scale for example [54]. Furthermore, it is worth noting that, even if no causality can be demonstrated here, the use of antidepressants was significantly higher in the group that perceived covid-19 discrimination. This effect should be interpreted with great caution. The findings of this study need to be further explored using a longitudinal study design to examine the prospective effect of perceived discrimination on health and emotional responses. This type of protocol would also facilitate the examination of the prospective effect of mental health or emotions on perceived discrimination. Finally, the proportion of individuals reporting discrimination in the COVISTRESS database seems to be surprisingly weak (135 out of 13429 participants, 1%). The number of missing data specifically for this question was particularly high (29% of the sample), suggesting that participants may have been embarrassed to answer questions. Instead, previous research has shown that about 20% of respondents reported discrimination (in U.S. and Indian samples; [22]) This surprisingly low proportion in the present study is likely due to underreporting, which is quite frequent in the case of personal social rejection [55,56], but also to a lower proportion of ethnic minorities in this database. Indeed, previous studies showed that non-

white participants are more likely to report covid-19 discrimination [22,26,28], as well as reporting more deleterious consequences on their mental health [28].

## Acknowledgments

"The COVISTRESS network is headed by Pr. Frédéric Dutheil (frederic.dutheil@uca.fr)–CHU Clermont-Ferrand, Occupational and Environmental Medicine, 58 rue Montalembert, 63000 Clermont-Ferrand, France. Members of the research group are Maëlys Clinchamps, Stéphanie Mestres, Cécile Miele, Valentin Navel, Lénise Parreira, Bruno Pereira, Karine Rouffiac–CHU Clermont-Ferrand, France; Yves Boirie, Jean-Baptiste Bouillon-Minois, Martine Duclos, Maria Livia Fantini, Jeannot Schmidt, Stéphanie Tubert-Jeannin–Université Clermont Auvergne / CHU Clermont-Ferrand, France; Mickael Berthon, Pierre Chausse, Michael Dambrun, Sylvie Droit-Volet, Julien Guegan, Serge Guimond, Laurie Mondillon, Armelle Nugier, Pascal Huguet–Université Clermont Auvergne, CNRS, LAPSCO, France; Samuel Dewavrin–WittyFit, France; Sébastien Couarraze, Louis Delamarre, Fouad Marhar–CHU Toulouse, France; Martial Mermillod–CHU Toulouse, France; Geraldine Naughton, Amanda Benson–Swinburne University, Australia; Claus Lamm–University of Vienna, Austria; Karen Gbaglo, Ministery of Health; Vicky Drapeau–Université de Laval, Canada; Raimundo Avilés Dorlhiac–Universidad Finis Terrae, Chile; Benjamin Bustos–Universidad de Los Andes, Chile; Gu Yaodong–Ningbo University, China; Haifeng Zhang–Hebei Normal University, China; Peter Dieckmann–Copenhagen Academy for Medical Education and Simulation (CAMES), Denmark; Julien Baker, Yanping Duan, Yang Gemma Gao, Yajun Wendy Huang, Jiao Jiao, Binh Quach, Chunqing Zhang, Hong Kong Baptist University, China; Hijrah Nasir, Indonesia; Perluigi Cocco, Rosamaria Lecca, Monica Puligheddu, Michela Figorilli, Università di Cagliari, Italia; Morteza Charkhabi, Reza Bagheri–University of Isfahan, Iran; Daniela Pfabigan–University of Oslo, Norway; Peter Dieckmann, University of Stavanger, Norway; Marek Zak, Tomasz Sikorski, Magdalena Wasik–Jan Kochanowski University of Kielce, Poland; Samuel Antunes, David Neto, Pedro Almeida–Ordem dos Psicólogos Portugueses, ISPA-Instituto Universitário, Portugal; Maria João Gouveia–ISPA-Instituto Universitário, Portugal; Pedro Quinteiro–William James Center for Research, ISPA-Instituto Universitário; Constanta Urzeala–UNEFS, Romania; Benoit Dubuis–UNIGE, Switzerland; Juliette Lemaignen–Fondation INARTIS, Switzerland; Kuan-Chou Chen, National Taiwan University of Sport, Taiwan; Andy Su-I Liu–University of Taipei, Taiwan; Foued Saadaoui, King Abdulaziz University, Tunisia; Ukadike C Ugbolue, University of the West of Scotland, United Kingdom; Keri Kulik–Indiana University of Pennsylvania, USA

## Author Contributions

**Conceptualization:** Michaël Dambrun, Eric Bonetto, Ladislav Motak, Jean-Baptiste Bouillon-Minois, Armelle Nugier, Maëlys Clinchamps, Frédéric Dutheil.

**Data curation:** Michaël Dambrun, Eric Bonetto, Ladislav Motak, Bruno Pereira, Maëlys Clinchamps, Frédéric Dutheil.

**Formal analysis:** Michaël Dambrun, Eric Bonetto, Ladislav Motak, Bruno Pereira.

**Investigation:** Michaël Dambrun, Julien S. Baker, Reza Bagheri, Foued Saadaoui, Hana Rabbouch, Marek Zak, Hijrah Nasir, Martial Mermillod, Yang Gao, Samuel Antunes, Ukadike Chris Ugbolue, Jean-Baptiste Bouillon-Minois, Armelle Nugier, Maëlys Clinchamps, Frédéric Dutheil.

**Methodology:** Michaël Dambrun, Eric Bonetto, Ladislav Motak, Bruno Pereira, Jean-Baptiste Bouillon-Minois, Maëlys Clinchamps, Frédéric Dutheil.

**Project administration:** Maëlys Clinchamps, Frédéric Dutheil.

**Resources:** Maëlys Clinchamps, Frédéric Dutheil.

**Software:** Bruno Pereira, Maëlys Clinchamps, Frédéric Dutheil.

**Supervision:** Frédéric Dutheil.

**Writing – original draft:** Michaël Dambrun, Eric Bonetto, Ladislav Motak, Frédéric Dutheil.

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
