## [Decision Letter · Decision Letter 0]

18 Aug 2022

PONE-D-22-19144Perceived discrimination based on the symptoms of covid-19, mental health and emotional responsesPLOS ONE

Dear Dr. Dambrun,

Thank you for submitting your manuscript to PLOS ONE. After careful consideration, we feel that it has merit but does not fully meet PLOS ONE’s publication criteria as it currently stands. Therefore, we invite you to submit a revised version of the manuscript that addresses the points raised during the review process.

We look forward to receiving your revised manuscript.

Kind regards,

Jianguo Wang, PhD

Academic Editor

PLOS ONE

Journal Requirements:

2. Please ensure that you have specified (1) whether consent was informed and (2) what type you obtained (for instance, written or verbal, and if verbal, how it was documented and witnessed). If your study included minors, state whether you obtained consent from parents or guardians. If the need for consent was waived by the ethics committee, please include this information.

 "The funders had no role in study design, data collection and analysis, decision to publish, or preparation of the manuscript". 

4. PLOS requires an ORCID iD for the corresponding author in Editorial Manager on papers submitted after December 6th, 2016. Please ensure that you have an ORCID iD and that it is validated in Editorial Manager. To do this, go to ‘Update my Information’ (in the upper left-hand corner of the main menu), and click on the Fetch/Validate link next to the ORCID field. This will take you to the ORCID site and allow you to create a new iD or authenticate a pre-existing iD in Editorial Manager. Please see the following video for instructions on linking an ORCID iD to your Editorial Manager account: https://www.youtube.com/watch?v=_xcclfuvtxQ.

5. One of the noted authors is a group or consortium "The COVISTRESS network". In addition to naming the author group, please list the individual authors and affiliations within this group in the acknowledgments section of your manuscript. Please also indicate clearly a lead author for this group along with a contact email address.

Reviewers' comments:

Reviewer's Responses to Questions

**Comments to the Author**

1. Is the manuscript technically sound, and do the data support the conclusions?

Reviewer #1: Partly

2. Has the statistical analysis been performed appropriately and rigorously? 

Reviewer #1: Yes

3. Have the authors made all data underlying the findings in their manuscript fully available?

Reviewer #1: Yes

4. Is the manuscript presented in an intelligible fashion and written in standard English?

Reviewer #1: No

5. Review Comments to the Author

Reviewer #1: This is a study on perceived discrimination based on the symptoms of covid-19, mental health and emotional responses. I have several comments on your interesting and important manuscript. This paper has potential, but it needs lots of additional work and major revisions. I am willing to propose it for publication once the following comments are addressed:

1. Study title: I suggest adding a study design in the title, for example survey/online questionnaire, or two designs based on COVISTRESS database (network) and online questionnaire. It is important to mention “online” and/or "database" in the title.

2. Keywords: In order to reach the wide audience for your work, I suggest adding more keywords, such as “online”, "databases" “questionnaire”, “emotional responses”, or something like these that include in your study.

3. Page 2, Abstract/Background: This section needs to be restructured. More information is needed to introduce why it is important to study this subject. Now, the section starts with “This study examines…”, and “…it was predicted…” (your hypothesis?). Before these sentences, background information on your study subject should be added.

4. Page 2, Abstract/Methods: The study period seems to be during the first wave of the pandemic. The year and months of your study period (questionnaires/survey conducted) should be mentioned in Abstract. Also, you may consider adding the statistical methods you used in this study.

5. Page 2, Abstract/Results: You mention “statistically controlled for” and “statistically significant” and also “relationship”. This section of Abstract should include statistical details found in your study. Please add more details, such as p-values and/or CIs, and/or correlations.

6. Page 3: The title “Introduction” is missing at the beginning of the first paragraph.

7. Introduction: Please use the numbered citations in brackets (not Riehm et al., 2012, for example). Thus, the list of references should be revised by using numbered references, appearing when mentioned for the first time in text, not in alphabetical order. Please check the PlosOne guidelines how to cite literature properly in this journal.

8. Different styles are used in terms of Covid-19, such as “covid-19”, “Covid 19”, “covid 19”, for example. Please use "Covid-19" consistently throughout the paper. Also, “Covid-19 related” and “Covid 19-related” are used. Please revise.

9. Citations in text: It is unnecessary to mention “e.g.”, for example (e.g., Miconi et al., 2020). Please revise and use numbered citations.

10. Methods, page 5: Respondents were from 44 countries. I think it would be important to list these countries, and mention in Abstract that 44 countries were included in this study. You may consider using a geographical map with all 44 countries placed on it, so that readers can see them at one glance. Also, the possible limitations (language and cultural differences may appear and affect the study results) related to several countries, should be discussed.

11. Results: Percentages are used when reporting results, such as 45.9%. I suggest using the absolute numbers in parentheses as well; % (n/N), so the readers can easily see the scale of your findings. Please check the PlosOne guidelines how to report percentages and results.

12. Page 9, Results: “We used Process 3.5 for SPSS (95% CI; Number of bootstrap samples = 5000)”. It is unnecessary to mention this twice, while the same sentence is already said in Methods. Please omit.

13. Table and Figure mentions in text: It is unnecessary to mention “see”, for example (see Table 2), just use (Table 2). Also, according to the journal guidelines, please use “Fig” instead of “Figure”.

14. Page 19, Table 1: The main participants’ characteristics include several details (variables) you found in this study. I think these should also be discussed in the Discussion section.

15. Since PlosOne has no copyediting services, I suggest checking up the English grammar and other copyediting issues thoroughly at this point of reviewing process. Please consult the journal guidelines, if necessary.

6. PLOS authors have the option to publish the peer review history of their article (what does this mean?). If published, this will include your full peer review and any attached files.

Reviewer #1: **Yes: **Samuli Pesälä

---

## [Author Response · Author response to Decision Letter 0]

15 Oct 2022

Reviewers' comments:

Reviewer #1: This is a study on perceived discrimination based on the symptoms of covid-19, mental health and emotional responses. I have several comments on your interesting and important manuscript. This paper has potential, but it needs lots of additional work and major revisions. I am willing to propose it for publication once the following comments are addressed:

1. Study title: I suggest adding a study design in the title, for example survey/online questionnaire, or two designs based on COVISTRESS database (network) and online questionnaire. It is important to mention “online” and/or "database" in the title.

Thank you for this suggestion. We added “- the international online COVISTRESS survey” in the title.

2. Keywords: In order to reach the wide audience for your work, I suggest adding more keywords, such as “online”, "databases" “questionnaire”, “emotional responses”, or something like these that include in your study.

We thank the reviewer for this suggestion. We added ‘emotional responses’ and “online COVISTRESS survey” to the keywords. 

3. Page 2, Abstract/Background: This section needs to be restructured. More information is needed to introduce why it is important to study this subject. Now, the section starts with “This study examines…”, and “…it was predicted…” (your hypothesis?). Before these sentences, background information on your study subject should be added.

The reviewer is right, we added the following sentence in the background section of our abstract p.2: ‘Despite its detrimental consequences for individuals' health, discrimination due to covid-19 symptoms has received little attention.’

4. Page 2, Abstract/Methods: The study period seems to be during the first wave of the pandemic. The year and months of your study period (questionnaires/survey conducted) should be mentioned in Abstract. Also, you may consider adding the statistical methods you used in this study.

Concerning the year and months of your study period, we added the following information in the abstract:

‘The study was conducted using an online questionnaire from January to June 2020 (Cov18istress network).’

and in the method section p.6:

‘Participants were extracted from the COVISTRESS database in January 2021 (the questionnaire was disseminated from January to June 2020; Couarraze et al., 2021).’

Concerning the statistical methods, we added the following sentence referring to the test of our main hypotheses: “Discriminated participants were compared to non-discriminated ones using ANOVA. A mediation analysis was conducted to examine the indirect effect of emotional responses in the relationship between perceived discrimination and self-reported mental health.”

5. Page 2, Abstract/Results: You mention “statistically controlled for” and “statistically significant” and also “relationship”. This section of Abstract should include statistical details found in your study. Please add more details, such as p-values and/or CIs, and/or correlations.

We would prefer to follow what is generally done in the journal (e.g., https://journals.plos.org/plosone/article?id=10.1371/journal.pone.0273813, https://journals.plos.org/plosone/article?id=10.1371/journal.pone.0273772), and thus avoid statistical details in the abstract.

6. Page 3: The title “Introduction” is missing at the beginning of the first paragraph.

We added this.

7. Introduction: Please use the numbered citations in brackets (not Riehm et al., 2012, for example). Thus, the list of references should be revised by using numbered references, appearing when mentioned for the first time in text, not in alphabetical order. Please check the PlosOne guidelines how to cite literature properly in this journal.

We followed your recommendation.

8. Different styles are used in terms of Covid-19, such as “covid-19”, “Covid 19”, “covid 19”, for example. Please use "Covid-19" consistently throughout the paper. Also, “Covid-19 related” and “Covid 19-related” are used. Please revise.

We thank the reviewer. We made the required changes. We used “Covid-19” through the paper.

9. Citations in text: It is unnecessary to mention “e.g.”, for example (e.g., Miconi et al., 2020). Please revise and use numbered citations.

We followed your recommendation.

10. Methods, page 5: Respondents were from 44 countries. I think it would be important to list these countries, and mention in Abstract that 44 countries were included in this study. You may consider using a geographical map with all 44 countries placed on it, so that readers can see them at one glance. Also, the possible limitations (language and cultural differences may appear and affect the study results) related to several countries, should be discussed.

We added the following information in the “participants” section: “The distribution of responses by continent was as follows: Europe 82.8%, America 8.5%, Africa 5,2%, Asia 3.4% (for more details, see Couarraze et al., 2021)”.

We also added a sentence in the discussion concerning the limitations (i.e. “This study was conducted in 44 countries using different languages. It is possible that there are cultural variations that have not been studied here and that may be studied when the covistress cohort will be of sufficient size in each country.”).

11. Results: Percentages are used when reporting results, such as 45.9%. I suggest using the absolute numbers in parentheses as well; % (n/N), so the readers can easily see the scale of your findings. Please check the PlosOne guidelines how to report percentages and results.

We relied on the way of reporting percentages used in recent articles published in PlosOne (e.g., https://journals.plos.org/plosone/article?id=10.1371/journal.pone.0273815). We would therefore prefer to keep these percentages as they are, unless the editor advises otherwise.

12. Page 9, Results: “We used Process 3.5 for SPSS (95% CI; Number of bootstrap samples = 5000)”. It is unnecessary to mention this twice, while the same sentence is already said in Methods. Please omit.

We thank the reviewer. We deleted this point.

13. Table and Figure mentions in text: It is unnecessary to mention “see”, for example (see Table 2), just use (Table 2). Also, according to the journal guidelines, please use “Fig” instead of “Figure”.

We made the required changes.

14. Page 19, Table 1: The main participants’ characteristics include several details (variables) you found in this study. I think these should also be discussed in the Discussion section.

We included this information to check that our two comparison groups are similar on these variables, and to identify possible covariates (which we already discuss). As it stands, there does not seem to be any major information emerging from this matching.

15. Since PlosOne has no copyediting services, I suggest checking up the English grammar and other copyediting issues thoroughly at this point of reviewing process. Please consult the journal guidelines, if necessary.

A co-author who is a native English speaker checked the grammar of this revised version.

With kind regard,

Pr. M. DAMBRUN and al.

---

## [Decision Letter · Decision Letter 1]

2 Dec 2022

Perceived discrimination based on the symptoms of covid-19, mental health, and emotional responses – the international online COVISTRESS survey

PONE-D-22-19144R1

Dear Dr. Dambrun,

We’re pleased to inform you that your manuscript has been judged scientifically suitable for publication and will be formally accepted for publication once it meets all outstanding technical requirements.

Kind regards,

Jianguo Wang, PhD

Academic Editor

PLOS ONE

Additional Editor Comments (optional):

Reviewers' comments:

Reviewer's Responses to Questions

**Comments to the Author**

1. If the authors have adequately addressed your comments raised in a previous round of review and you feel that this manuscript is now acceptable for publication, you may indicate that here to bypass the “Comments to the Author” section, enter your conflict of interest statement in the “Confidential to Editor” section, and submit your "Accept" recommendation.

Reviewer #1: All comments have been addressed

2. Is the manuscript technically sound, and do the data support the conclusions?

Reviewer #1: Yes

3. Has the statistical analysis been performed appropriately and rigorously? 

Reviewer #1: Yes

4. Have the authors made all data underlying the findings in their manuscript fully available?

Reviewer #1: Yes

5. Is the manuscript presented in an intelligible fashion and written in standard English?

Reviewer #1: Yes

6. Review Comments to the Author

Reviewer #1: Authors have addressed all the comments properly, and the paper has significantly improved. I suggest that this manuscript is accepted.

7. PLOS authors have the option to publish the peer review history of their article (what does this mean?). If published, this will include your full peer review and any attached files.

Reviewer #1: **Yes: **Samuli Pesälä

---

## [Editor Report · Acceptance letter]

13 Dec 2022

PONE-D-22-19144R1 

Perceived discrimination based on the symptoms of covid-19, mental health, and emotional responses – the international online COVISTRESS survey 

Dear Dr. Dambrun:

I'm pleased to inform you that your manuscript has been deemed suitable for publication in PLOS ONE. Congratulations! Your manuscript is now with our production department. 

Kind regards, 

on behalf of

Dr. Jianguo Wang 

Academic Editor

PLOS ONE